# Adolescent Alcohol Drinking Renders Adult Drinking BLA-Dependent: BLA Hyper-Activity as Contributor to Comorbid Alcohol Use Disorder and Anxiety Disorders

**DOI:** 10.3390/brainsci7110151

**Published:** 2017-11-14

**Authors:** Mahsa Moaddab, Elizabeth Mangone, Madelyn H. Ray, Michael A. McDannald

**Affiliations:** Department of Psychology, Boston College, Chestnut Hill, MA 02467, USA; mangonee9@gmail.com (E.M.); raymd@bc.edu (M.H.R.)

**Keywords:** ethanol, chronic, intermittent, voluntary, rat, amygdala, fear

## Abstract

Adolescent alcohol drinking increases the risk for alcohol-use disorder in adulthood. Yet, the changes in adult neural function resulting from adolescent alcohol drinking remain poorly understood. We hypothesized that adolescent alcohol drinking alters basolateral amygdala (BLA) function, making alcohol drinking BLA-dependent in adulthood. Male, Long Evans rats were given voluntary, intermittent access to alcohol (20% ethanol) or a bitter, isocaloric control solution, across adolescence. Half of the rats in each group received neurotoxic BLA lesions. In adulthood, all rats were given voluntary, intermittent access to alcohol. BLA lesions reduced adult alcohol drinking in rats receiving adolescent access to alcohol, but not in rats receiving adolescent access to the control solution. The effect of the BLA lesion was most apparent in high alcohol drinking adolescent rats. The BLA is essential for fear learning and is hyper-active in anxiety disorders. The results are consistent with adolescent heavy alcohol drinking inducing BLA hyper-activity, providing a neural mechanism for comorbid alcohol use disorder and anxiety disorders.

## 1. Introduction

Alcohol is widely used among adolescents and young adults [1]. Adolescent alcohol drinking, particularly heavy drinking, is associated with increased risk for the development of alcohol use disorder in adulthood [2,3]. Consistent with observations in people, recent studies in rodents have shown that voluntary alcohol drinking in adolescence can increase alcohol drinking [4] or permit alcohol access to appetitive learning [5] in adulthood. While adolescents appear to be potentially more vulnerable to the effects of alcohol [6], the precise neural mechanisms that mediate this vulnerability have yet to be determined.

Our laboratory recently demonstrated that adolescent heavy alcohol drinking impairs rapid fear discrimination in adulthood [7]. Specifically, adolescent heavy drinkers are unable to rapidly reduce fear to safety cues, showing excessive fear. This impairment is likely the result of adolescent alcohol drinking, altering the function of brain regions critical to fear, most notably the basolateral amygdala (BLA) [8,9,10,11,12,13]. We hypothesize that adolescent drinking alters BLA function outside of fear. Specifically, we hypothesize that alcohol drinking, established in adolescence, depends on the BLA in adulthood. If confirmed, this would uncover the BLA as a key node of dysfunction for adult fear, *and* adult alcohol drinking, resulting from adolescent heavy alcohol drinking. To test this hypothesis, rats were given voluntary, intermittent access to alcohol or a bitter, isocaloric control solution throughout adolescence. Half of the rats in each group then received bilateral, neurotoxic BLA lesions. Following surgery, and in adulthood, all rats were given voluntary access to alcohol. This allowed us to determine the role of the BLA in adult alcohol drinking, established in adolescence or adulthood.

## 2. Materials and Methods

### 2.1. Subjects

Forty-eight male, Long Evans rats were obtained from Taconic Laboratories on postnatal day 21 ± 3. Rats were singly housed, and maintained on a 12 h light cycle (lights off at 6:00 p.m.) in a home cage. Food and water were freely available at all times. Procedures adhered to the Institute for Laboratory Animal Research Guide for the Care and Use of Laboratory Animals, and were approved by the Boston College Animal Care and Use Committee (Protocol # 2018-002, Approved 25 July 2017).

### 2.2. Apparatus

All experiments took place in the home cage. Experimental solutions were placed in experimental bottles (50 mL centrifuge tubes with rubber stoppers and ball bearing sipper tubes [14]) and secured to the wire cage top with a large binder clip. The location of the bottle alternated from the left to the right side of the cage top each session. Rat chow and standard water bottles were always present on the home cage. The standard water bottles were designed to be slightly leaky. This ensured that any drinking of experimental solutions was not the result of dehydration.

### 2.3. Adolescent Voluntary Intermittent Access to Alcohol

Starting on postnatal day 23 ± 3, half of the rats (24/48) were given voluntary intermittent access to alcohol (20% ethanol, *v/v*). Voluntary intermittent access was based off a procedure developed by Simms and colleagues [15] and was identical to access previously used by our laboratory [5,7]. Experimental bottles were filled with ~40 mL of 20% ethanol, and placed on the cage at 10:30 a.m. on Sunday, Tuesday, and Thursday. Experimental bottles were removed at 10:30 a.m. on Monday, Wednesday and Friday. Rats received voluntary intermittent access to alcohol until postnatal day 56 ± 3, resulting in a total of 16 access sessions. We chose to end the procedure on this day in order to provide access over the entirety of adolescence, but to end before adulthood, typically described as beginning on postnatal day 60 at the earliest [16]. The remaining half (24/48) received the same procedure, but were given voluntary access to experimental bottles containing quinine-adulterated DuoCal^®^ (QuAD; 0.001% quinine, *w/v*, 28% DuoCal^®^, *w/v*) to serve as a bitter, isocaloric control for alcohol.

### 2.4. Dependent Measures

Experimental bottles were weighed (g) prior to, and following, 24 h access to solutions. Rats were weighed (kg) immediately following experimental bottle removal. Alcohol and QuAD drinking are reported in grams (solution consumed)/kg (body weight) per 24 h (total time given access): g/kg/24 h. Because the alcohol solution was ~80% water, ethanol drinking levels were lower than alcohol drinking levels. Ethanol drinking (g/mL of 20% ethanol) was determined by multiplying alcohol drinking by a constant factor (0.162) [17,18]. Importantly, this transformation did not change the underlying distribution of drinking. Identical statistical results were obtained if alcohol drinking or ethanol drinking data were analyzed (Appendix A).

### 2.5. Surgical Procedures

Surgery began one day after the conclusion of voluntary intermittent access (postnatal day 57 ± 3). Rats from each drinking condition were split into two groups: Control or BLA lesion (BLAx), equating for means and standard deviations in drinking. BLA lesions were made via stereotaxic surgery, using infusions of N-methyl-D-aspartate (NMDA) in phosphate buffer solution (20 μg/μL, *w/v*). Each BLAx rat received four total infusions of the following volumes, at the following coordinates: 0.133 μL was infused bilaterally at: anterior-posterior (AP), −2.3 mm; medial-lateral (ML), ±4.60 mm; dorsal-ventral (DV), −7.7 mm; and 0.1 μL of NMDA was infused bilaterally at: AP, 2.3 mm; ML ±4.60 mm; DV, −7.6 mm. Two DV coordinates were used per hemisphere, in order to more completely lesion the dorsal–ventral extent of the BLA. Controls received identical treatment, but no infusion was administered. Stereotaxic surgery was performed under isoflurane anesthesia (1–5% in oxygen) with aseptic technique. Carprofen (i.p., 5 mg/kg) was used for post-operative analgesia.

### 2.6. Adult Voluntary Intermittent Access to Alcohol

Twenty days post-surgery (postnatal day 77 ± 3) all rats received eight sessions of voluntary intermittent access to alcohol, exactly as described above. Food and water were always available. Drinking (g) and body weight (kg) were recorded as described above.

### 2.7. Histological Procedures

After adult voluntary access to alcohol, all rats were anesthetized and perfused with 0.9% saline. Brains were removed, post-fixed overnight in 10% formalin, then fixed 24 h in the same formalin solution plus 10% sucrose. Brains were sliced on a freezing microtome and 40 µm coronal sections through the BLA were collected. Sections were mounted, Nissl stained and examined with a light microscope.

### 2.8. Statistics

Alcohol drinking, by body weight (g/kg/24 h), was analyzed with SPSS (Version 23.0, IBM, Armonk, NY, USA) using analysis of variance (ANOVA). Post-hoc comparisons for alcohol rats were performed with dependent samples, two-tailed *t*-tests. The square of Pearson’s correlation coefficient (*R*^2^) was used to determine the relationship between pre-surgery and post-surgery drinking, in alcohol-Control and alcohol-BLAx individuals. The sign test was used to determine pre-surgery to post-surgery shifts in drinking in each alcohol group. Post-hoc comparisons for QuAD rats were performed with independent samples, two-tailed *t*-tests. Line graphs are plotted as mean + SEM for intact rats and mean − SEM for BLAx. This was done to best visualize statistical effects that would otherwise be obscured by plotting mean ± SEM for both groups. For all graphs, the mean + SEM value exactly mirrored the corresponding mean − SEM value. In all cases, *p* < 0.05 was considered significant.

## 3. Results

### 3.1. Histology

We outlined neurotoxic damage (cell loss and gliosis) for each individual at five distances from bregma (−2.16, −2.52, −2.92, −3.24, and −3.60), based on the Watson and Paxinos atlas [19]. The outline for each individual was made 5% opaque, and then all individuals were stacked for each bregma level. The result is shown in Figure 1A. Darker areas indicate regions of high overlap and consistent neurotoxic damage. Lighter areas indicate regions of low overlap and less consistent damage. Neurotoxic damage was apparent at all BLA levels, with the most consistent damage found in the anterior and posterior basal nuclei (>80% damage), as well as in the ventromedial and ventrolateral lateral nuclei (>70% damage). Controls showed no evidence of neurotoxic damage. Representative micrographs of Nissl-stained sections from a Control (Figure 1B) and a BLAx rat (Figure 1C) are shown. No differences in location or extent of BLA lesions were observed for alcohol compared to QuAD rats.

### 3.2. Alcohol and QuAD Drinking in Adolescence

Initially, rats were divided into two groups and given access to alcohol (*n* = 24) or QuAD (*n* = 24) over adolescence. At the end of adolescent access, each group was divided into control and BLAx conditions, equating for drinking. While the QuAD solution was exactly matched for caloric content, significantly greater QuAD drinking was observed. In support, the ANOVA for drinking (g/kg/24 h) with group (alcohol vs. QuAD), lesion (Control vs. BLAx) and session (1–16) as factors, found significant main effects for group and session (*Fs* > 7, *ps* < 0.01) and for the group x session interaction (*F*_15,675_ = 7.01, *p* < 0.01). Mean ± SEM QuAD drinking over the 16 sessions was 102.7 ± 12.2, while alcohol drinking was 7.3 ± 1.0. Importantly, ANOVA found no main effect or interaction of lesion (*Fs* < 0.6, *ps* > 0.9), indicating the groups were appropriately balanced for drinking. Because drinking levels were so different between alcohol and QuAD rats, and because we were primarily interested in changes in adult alcohol drinking levels produced by BLA lesions following adolescent alcohol access, we analyzed drinking in the two groups separately.

### 3.3. Adult Alcohol Drinking Established in Adolescence Depends on the BLA

Consistent with previous findings from our lab [5,7], alcohol drinking by bodyweight was highest during initial access, declined and then stabilized over the 16 sessions of voluntary access (Figure 2A, sessions 1–16). In support, the ANOVA for alcohol drinking (g/kg/24 h) with lesion (control vs. BLAx) and session (1–16) as factors, found a significant main effect of session (*F*_15,330_ = 6.43, *p* < 0.01). However, when the same ANOVA was confined to the final eight sessions (9–16), only a trend towards session significance was found (*F*_15,154_ = 1.90, *p* = 0.07). Inclusion of the lesion condition in this analysis ensured that differences in drinking did not exist between control and BLAx rats prior to surgery; all rats had the BLA intact during sessions 1–16.

In order to determine the effect of BLA lesions on alcohol drinking established in adolescence, we analyzed the eight voluntary sessions prior to, and following surgery. Pre-surgery drinking had stabilized during these eight sessions, as indicated by ANOVA, revealing no main effect of session. Further, examining the eight sessions on either side of surgery assured us that the analysis of drinking was balanced (identical number of pre-surgery and post-surgery sessions) and complete (analyzing the full post-surgery drinking data collected).

BLA lesions significantly reduced adult alcohol drinking established in adolescence. In support, the ANOVA for alcohol drinking (g/kg/24 h) with lesion (Control vs. BLAx) and session (9–24) as factors, found a significant lesion x session interaction (*F*_15,330_ = 2.05, *p* = 0.01). The interaction was the result of BLAx rats reducing alcohol drinking following surgery, while control rats continued voluntary alcohol drinking (Figure 2B). The dependent samples, two-tailed *t*-test for eight-session mean drinking found that while control rats showed no difference in pre- and post-surgery drinking (*p* > 0.05), BLAx rats showed a significant reduction from pre- to post-surgery (*p* < 0.05). Post-surgery alcohol drinking levels were 6.33 ± 2.04 g/kg/24 h for controls, and 3.85 ± 1.42 g/kg/24 h for BLAx rats.

We were next interested in determining if individual variation in adolescent alcohol drinking influenced post-surgery, adult drinking. To do this, we compared the eight-session mean of pre- and post-surgery drinking for all individuals in each group (Figure 3A,B). In controls, the majority of individuals showed low alcohol drinking pre-surgery, with a smaller number of individuals showing high alcohol drinking. Interestingly, there was a strong correlation between pre- and post-surgery drinking across all control rats (*R*^2^ = 0.79, *p* < 0.01). Rats showing low drinking prior to sham surgery appeared to decrease drinking following sham surgery (Figure 3B). However, control rats showing high drinking prior to sham surgery continued to show high drinking or even increased drinking following sham surgery. As expected, there was no group-wide shift towards greater or lesser drinking following sham surgery in control rats (sign test, *p* = 0.39).

Similarly, the majority of rats in the BLAx group showed low alcohol drinking prior to surgery, with a smaller number of individuals showing high drinking. There was a strong correlation between pre-surgery and post-surgery drinking (*R*^2^ = 0.91, *p* < 0.01). BLAx rats showing low drinking prior to surgery decreased following surgery, as did controls (Figure 3D). However, BLAx rats showing high drinking prior to surgery, also decreased drinking following surgery. In contrast to controls, there was a significant, group-wide shift towards reduced drinking following neurotoxic BLA lesions (sign test, *p* < 0.01; Figure 3C,D). BLA lesions reduced adult alcohol drinking, irrespective of adolescent low or high alcohol drinking.

Examination of the individual data suggests that the effect of BLA lesions in reducing alcohol drinking was most apparent in rats showing high alcohol drinking prior to surgery. In order to visualize the contribution of pre-surgery drinking level, we plotted the same drinking data from Figure 2, only now separately for the top four (Figure 4A,B represent high) and bottom eight (Figure 4C,D represent low) alcohol drinkers in each group. Plotted in this way, the session x lesion interaction observed across all rats appeared to be largely the result of BLA lesions reducing drinking in adolescent high alcohol drinkers.

### 3.4. Adult Alcohol Drinking Established in Adulthood Does not Depend on the BLA

To determine if the BLA is specifically necessary when alcohol drinking is established in adolescence, we analyzed post-surgery alcohol drinking in control and BLAx rats receiving QuAD access in adolescence (Figure 5). A full ANOVA for pre- and post-surgery drinking was not informative, due to the overwhelming effect of session, driven by high QuAD drinking. The ANOVA for the eight post-surgery sessions found a main effect of session (*F*_7,154_ = 8.04, *p* < 0.01) but no effect of lesion or the lesion x session interaction (*Fs* < 2, *ps* > 0.2). Post-surgery alcohol drinking levels were 5.23 ± 1.12 g/kg/24 h for controls and 3.96 ± 0.60 g/kg/24 h for BLAx rats. Even when only the first-session drinking was compared, using an independent sample, two-tailed test, no significant effect was found (Figure 5B). This brief increase in drinking, restricted to the first post-surgery session, was most likely due to residual drinking supported by the non-alcoholic, control solution in adolescence. Indeed, when only the final four sessions of post-surgery drinking were measured across all groups, the highest alcohol drinking was observed in alcohol-intact rats: 6.51 ± 2.09 g/kg/24 h, with lesser and equivalent drinking observed in the remaining groups: alcohol-BLAx: 3.89 ± 1.70 g/kg/24 h, QuAD-intact: 4.31 ± 1.26 g/kg/24 h, and QuAD-BLAx: 3.73 ± 0.83 g/kg/24 h. The results support the hypothesis that alcohol drinking established in adolescence, but not adulthood, is BLA-dependent.

## 4. Discussion

Our results demonstrate that the BLA plays a minimal role in voluntary alcohol drinking established in adulthood. This is directly in line with a previous study showing that BLA lesions had no impact on alcohol drinking in alcohol-naïve adults [20]. However, it is clear from our results that a history of adolescent alcohol drinking—particularly heavy drinking—alters BLA function in adulthood. BLA lesions markedly reduced adult alcohol drinking in rats receiving adolescent alcohol access. Interestingly, the effect of BLA lesions was not complete. The highest adolescent alcohol drinking rats to receive a BLA lesion showed the highest adult drinking. Furthermore, the relationship between adolescent and adult alcohol drinking was maintained even in BLA-lesioned rats: low drinking adolescents were low drinking adults and high drinking adolescents were (relatively) high drinking adults. This could be due to incomplete BLA lesions. For example, the dorsal subregion of the lateral amygdala was not consistently damaged in this study. An account we find more plausible is that adolescent alcohol drinking alters the adult function of a distributed neural circuit. Within this neural circuit, and through adolescent alcohol drinking, the BLA may control the ‘gain’ on adult alcohol drinking. Removal of the BLA then does not abolish the relationship between adolescent and adult drinking, but results in less adult drinking than if the BLA were intact.

In support of a network view, a host of brain regions have been shown to be altered by adolescent alcohol drinking [21]. A recent study by Liu and Crews gave adolescent rats involuntary alcohol experience over a time span roughly similar to our procedure. In adulthood, rats received an alcohol challenge, and neural activity was assessed with the immediate early gene, *c-fos*. They found widespread, alcohol-induced changes in c-fos that were specific to rats receiving adolescent alcohol exposure [22]. They identified brain regions that had previously been shown to be targets of adolescent alcohol, including the central amygdala [23], ventral tegmental area [24,25], nucleus accumbens [24,25,26], prefrontal cortices [27], orbitofrontal cortex [28], as well as the BLA. Rather than overturn these findings, the present study strongly suggests that—along with these regions—the BLA is a critical node in a distributed neural circuit targeted by adolescent heavy alcohol drinking. This is consistent with anatomical studies showing the BLA densely projects to the accumbens, prefrontal cortices, orbitofrontal cortex, and central amygdala [29,30].

The results pose further questions for the role of the BLA in adolescent to adult alcohol drinking. The first concerns the timeline of BLA involvement. Here, we gave BLA lesions following extensive access to alcohol in adolescence. By observing a reduction in adult alcohol drinking, we conclude that adolescent alcohol-induced changes in neural function normally occur in a network that includes the BLA. However, if adolescent heavy alcohol drinking targets the BLA—making adult alcohol drinking BLA-dependent—then BLA lesions in early adolescence may prevent the emergence of adult heavy alcohol drinking altogether. Alternatively, it may be that in the absence of the BLA, alcohol drinking targets different brain regions. Thus, adult alcohol drinking could still be observed, but would now be BLA-independent. The second question concerns the BLA neuron-type affected by alcohol drinking. The BLA is primarily composed of glutamatergic, pyramidal output neurons, whose activity is regulated by local, gamma-aminobutyric acid (GABA) interneurons. BLA output is ultimately the result of both populations, and determining which neuron-type is impacted by adolescent drinking would be of great value. As we will discuss below, there is evidence that the function of both these neuron-types is altered by alcohol drinking.

Finally, the current results are particularly interesting in light of the established role for the BLA in fear, and in light of a previous study from our laboratory showing that adolescent heavy alcohol drinking impairs rapid fear discrimination in adulthood [7]. That study used the same voluntary access procedure; only control rats received water. Following voluntary access, adolescent heavy drinkers were identified as rats consuming greater than 10 g/kg/24 h of 20% ethanol—levels typically found in rats bred to consume alcohol [31,32]. Moderate alcohol drinkers consumed well under 5 g/kg/24 h. After an abstinence period, all rats were assessed in a fear discrimination procedure, in which three cues predicted foot shock with unique probabilities: danger (*p* = 1.00), uncertainty (*p* = 0.25) and safety (*p* = 0.00). We were curious how quickly fear discrimination could be detected following cue onset. We found that control rats and adolescent moderate alcohol drinkers were very quick, showing discriminative fear less than one second following cue onset. Adolescent heavy drinkers were markedly impaired in rapid fear discrimination, and were specifically unable to rapidly reduce fear to the safety cue [7]. When fear discrimination was assessed later in cue presentation (the final 1 s of 10 s cues), heavy drinkers showed performances equivalent to that of controls, demonstrating they were not generally unable to discriminate. Thus, adolescent heavy alcohol drinking appears to specifically disrupt rapid fear discrimination—resulting in excessive fear to safe cues.

In light of these findings, and the observation that adolescent heavy alcohol drinking increases the risk for alcohol use disorder and anxiety disorders, we suggest that adolescent heavy alcohol drinking results in BLA hyper-activity in adulthood, and perhaps for the lifespan. We will briefly discuss the established roles for the BLA in fear, alcohol drinking and known effects of alcohol on BLA function. We then present plausible scenarios for BLA hyper-activity due to adolescent alcohol drinking contributing to excessive fear and alcohol drinking.

### 4.1. BLA Contribution to Fear

The BLA is perhaps best known for its role in Pavlovian fear conditioning [10]. In a simple fear conditioning procedure, a novel cue is repeatedly paired with aversive foot shock, an intrinsically aversive event. Even with a limited number of pairings, the cue will acquire a variety of associative properties [33,34,35], and will control autonomic [36] and behavioral responses [8,37], including freezing. It has been repeatedly demonstrated the BLA is necessary for fear learning. Neurotoxic BLA lesions impair the acquisition and expression of fear [11,12,36,38,39,40,41]. Lesion findings are buoyed by pharmacological inactivation studies, which show that interfering with BLA activity only at the time of learning, or expression, is sufficient to impair fear [42,43,44]. Even more, robust immediate early gene expression is observed in the BLA, following exposure to fearful cues [45,46,47,48,49].

BLA single units show changes in firing, indicative of fear learning. BLA units are minimally responsive to novel cues, but through pairing with foot shock, come to elicit robust activity to the same cue [50,51,52]. Fear-learning related changes in activity have been ascribed to glutamatergic, pyramidal neurons, which provide the majority of the BLA output. The prevailing view is that fear acquisition, and fear-related firing by BLA neurons, are the product of long-term potentiation of cue-specific sensory inputs onto BLA pyramidal neurons [53,54]. Pairing a cue with shock strengthens the synaptic weights of only the specific sensory inputs associated with foot shock. Future presentation of the cue is then sufficient to elicit robust activity in single BLA neurons—thus, driving fear. It is now clear that local, GABAergic interneurons [55] are critical for fear learning and expression [51]. While the exact mechanisms are still being uncovered [56], it is likely that pyramidal neuron output is due, in part, to disinhibition via GABAergic interneurons. That is, decreased firing of BLA GABAergic interneurons (inhibition) permits increased firing of BLA pyramidal neurons. Normal expression of fear then depends on a balance of local, GABAergic inhibition and projecting glutamatergic excitation [57,58].

Just as BLA activity is considered to underlie normal fear learning, BLA hyper-activity is considered to be a key feature of anxiety disorders. In healthy people, increased amygdala activity (as measured by functional magnetic resonance imaging) is routinely observed in response to angry or fearful faces [59,60,61]. In people with generalized social anxiety disorder, the amygdala is hyper-active to angry or fearful faces, showing a response beyond that of age-matched, healthy controls [62]. Amygdala hyper-activity is not unique to generalized social anxiety disorder, but has been shown to be a key feature of a variety of disorders including, but not limited to, social anxiety disorder [63], specific phobias [64] and post-traumatic stress disorder [65]. A particularly compelling meta-analysis of imaging studies on brain-wide BOLD responses to a variety of emotional images, found consistent amygdala hyper-activity across anxiety disorders [66]. While it cannot be concluded that amygdala hyper-activity plays a *causal* role in anxiety disorders, at the very least it must be a key symptom or feature. We will return to this observation when we consider scenarios for the disruption of BLA function by adolescent drinking.

Additionally, it has been shown that individuals with post-traumatic stress disorder (PTSD) are impaired in safety learning [67]. In healthy individuals, cues predicting the absence of danger (safety) are capable of suppressing fear responses to cues predicting danger. The ability of safety cues to suppress fear is markedly impaired in individuals with PTSD. While the precise neural dysfunction is yet to be uncovered, it is likely that in PTSD, safety signals originating within [52,68] or outside the BLA [69,70] fail to regulate BLA responses to danger; resulting in an inappropriate fear to safe events. BLA hyper-activity is then likely to exaggerate fear responses to emotional stimuli and danger, yet can also produce inappropriate fear responses to innocuous stimuli and safety.

### 4.2. BLA Contribution to Alcohol-Related Behavior

In comparison to fear conditioning, considerably less is known of the BLA contribution to alcohol-related behaviors. However, studies have demonstrated that the BLA is integral to the process by which alcohol-associated cues or contexts acquire motivational properties. The BLA also plays a role in more fundamental alcohol-related behaviors, such as alcohol seeking and drinking. For example, blocking BLA activity [71] or BLA glutamate receptors [72,73] blocks the ability of an alcohol-associated cue to invigorate actions specific to obtaining alcohol. The BLA is critical to the renewal of motivational properties of alcohol-associated cues extinguished in another context [74,75]. In line with these observations, alcohol cues that reinstate extinguished alcohol seeking also increase glutamate release [76], and immediate early gene expression [77] in the BLA. Alcohol seeking depends, in part, on the modulation of BLA pyramidal neurons [78,79]. BLA projections to the nucleus accumbens shell-regulate both learned alcohol behaviors and voluntary alcohol drinking [80]. The BLA is then essential for driving a variety of alcohol-associated behaviors, with the general finding that blocking or inhibiting BLA activity reduces alcohol seeking and alcohol-related behaviors (except [80]).

None of these studies explicitly gave rats adolescent access to alcohol. This would appear to conflict with our hypothesis that the BLA is not critical to alcohol-associated behaviors in alcohol-naïve individuals. However, in all of these studies, rats were given chronic access to alcohol, well before testing of alcohol drinking, seeking or other learned behaviors. Further, our adolescent drinking procedure gave rats alcohol access nearly up until adulthood (postnatal day 56 ± 3). It is possible that while adolescent alcohol drinking may particularly impact BLA function, chronic drinking initiated later in life might produce similar effects.

### 4.3. Impact of Alcohol Drinking on BLA Function

In addition to regulating alcohol-related behaviors, the function of the BLA itself is altered by alcohol experience. For example, chronic alcohol drinking enhances glutamate transmission in the BLA, via pre- and post-synaptic mechanisms [81,82,83,84]. Chronic drinking simultaneously decreases local, GABA inhibitory input onto BLA pyramidal neurons, via the disruption of CB1 receptors [85]. Chronic alcohol drinking, and subsequent withdrawal also alter the subunit composition of BLA GABA receptors [86,87,88]. Voluntary alcohol drinking in adolescence, as opposed to adulthood, results in higher ΔFosB expression in the BLA [89]. ΔFosB is a transcription factor associated with drug-induced neural plasticity [90] and has been suggested as a neural marker for vulnerability to alcohol use disorder [89,91]. Overall, these findings suggest that alcohol drinking affects BLA function in a manner that would ultimately lead to hyper-activity or an increased BLA output. This is consistent with the broader proposal that chronic alcohol use may result in a hyper-glutamatergic state [92].

Finally, and perhaps most pertinent, there is evidence that alcoholism (now referred to as alcohol use disorder) is associated with altered gene expression in human BLA. In this study, post-mortem BLA samples were analyzed from adult male alcoholics and age-matched controls. While genes of many categories differed between the two groups, most striking was differential expression of genes pertaining to the glutamate system [93]. Indeed, the observed glutamate gene alterations would have *increased* glutamate tone, resulting in a hyper-active state. While this is only a single, post-mortem result, this is directly in line with the idea of chronic alcohol use resulting in BLA hyper-activity.

### 4.4. Proposed Alteration of BLA Function by Adolescent Alcohol Drinking

We propose that the BLA plays a selective role in fear in alcohol-naïve adults, or adults with an adolescent history of moderate alcohol drinking. In these individuals, BLA pyramidal and GABA populations are either non-responsive, or minimally-responsive, during alcohol drinking (Figure 6A). Instead, BLA pyramidal and GABA populations are selectively responsive to fear cues (selective with respect to fear and alcohol, the BLA contributes to a variety of motivated behaviors not discussed here [94,95,96,97]). Specifically, BLA neurons rapidly scale activity to reflect the probability of shock (danger > uncertainty > safety). The relative activity of pyramidal and GABA populations is balanced so that the BLA output produces appropriate levels of fear. Critically, BLA pyramidal activity to the safety cue is low or absent, permitting the expression of low/no fear.

Adolescent heavy alcohol drinking co-opts specific populations of BLA pyramidal and GABA interneurons. In one scenario, adolescent heavy alcohol drinking co-opts unique populations that did not previously contribute to fear (Figure 6B). In a second scenario, heavy adolescent drinking co-opts existing pyramidal and GABA populations for fear (Figure 6C). Most critically, adolescent heavy alcohol drinking inflates BLA output for both fear and alcohol—rendering the BLA hyper-active. In both scenarios, this is achieved through a combination of weakening local inhibitory control by GABA, yet also exaggerating the output of pyramidal neurons. The net result is that in adults with heavy drinking histories, the BLA is hyper-active, driving excessive fear—particularly at onset or initial encounter—and also driving excessive drinking.

In our proposal, it not only becomes clear why adolescent heavy alcohol drinking increases the lifetime risk for both alcohol use disorder and anxiety disorders, but potentially why there is such strong comorbidity between the two disorders—both feature amygdala hyper-activity. In support of our proposal, it has been shown that not only is the amygdala responsive to alcohol cues in people with alcohol use disorder, but that amygdala activation predicts the degree of alcohol dependence and the failure of control over drinking [98].

Immediately, we can think of two strong tests of our hypothesis. The first is a functional magnetic resonance imaging experiment, in which two groups of individuals—adolescent heavy/binge alcohol drinkers and age/background matched individuals—would be scanned during procedures assessing blood-oxygen-level dependent responses to alcohol or alcohol-associated cues [99] and also during fear discrimination assessing danger, uncertainty and safety [100]. We predict that the BLA would be hyper-responsive to alcohol or associated cues, as well to the onset of cues in fear discrimination—particularly to the safety cue. Within our hypothesis, this experiment would begin to determine whether Scenario #1 (independent signaling of alcohol and fear) or Scenario #2 (joint signaling), was more likely. For example, finding significant voxel-level correlations between BLA responses to alcohol, and cues in fear discrimination, would support the hypothesis that adolescent heavy alcohol drinking co-opts BLA populations underlying fear. The second would be identical in nature, only now in wild-type Long Evans rats. Moderate/heavy drinkers would be identified in adolescence. BLA single-unit activity would be recorded during fear discrimination and alcohol drinking/seeking in adulthood. This would allow us to determine if single BLA neurons are hyper-active to cues in fear discrimination and during alcohol drinking—and the degree to which firing is related. Studies of this nature are now underway in our laboratory.

## 5. Conclusions

Here we have shown that the BLA is critical to adult alcohol drinking established in adolescence, a role that is most apparent in adolescent heavy alcohol drinkers. This finding, combined with known roles for the BLA in fear, alcohol-related behaviors and alcohol-induced alteration of BLA function, points to BLA-hyperactivity as a contributor to the comorbidity of alcohol use and anxiety disorders. While the BLA is almost certainly not the only brain region contributing to comorbid alcohol and anxiety [23,101,102], the current study suggests that greater attention should be paid to its contribution. Comorbid alcohol use disorder and anxiety disorders is unfair in the extreme. Efforts to quit drinking are significantly more likely to fail in those with comorbid anxiety disorders [103,104]. However, comorbidity may come with a silver lining. If common neural mechanisms, such as BLA hyper-activity, underlie alcohol use disorder and anxiety disorders, then treatments to restore neural function in one disorder may restore function in the other [105].

## Figures and Tables

**Figure 1 brainsci-07-00151-f001:**
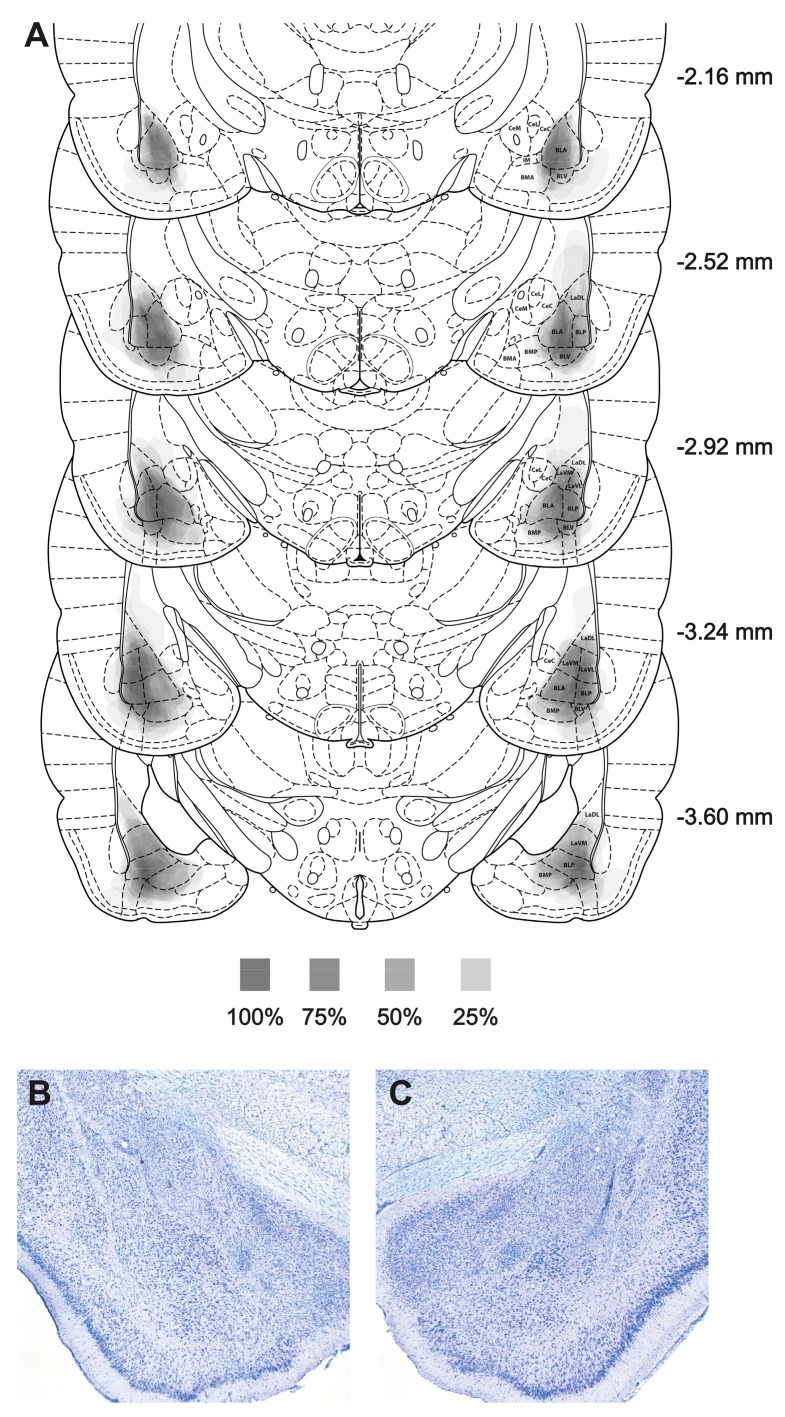
Histology. (**A**) The extent of neurotoxic basolateral amygdala (BLAx) lesions across five coronal planes is shown, and the posterior distance from bregma (millimeters) indicated. Each individual’s lesion was drawn (*n* = 24), shaded at 5% opacity using Adobe Photoshop (CS6) and stacked. Darker areas indicate greater overlap between individual lesions, and thus, greater damage. Representative Nissl images showing (**B**) Control with BLA intact and (**C**) BLAx with a neurotoxic lesion. BLA, basolateral amygdaloid nucleus, anterior part; BLP, basolateral amygdaloid nucleus, posterior part; BLV, basolateral amygdaloid nucleus, ventral part; BMA, basomedial amygdaloid nucleus, anterior part; BMP, basomedial amygdaloid nucleus, posterior part; CeC, central amygdaloid nucleus, capsular part; CeL, central amygdaloid nucleus, lateral part; CeM, central amygdaloid nucleus, medial part; IM, intercalated amygdaloid nucleus, main part; LaDL, lateral amygdaloid nucleus, dorsolateral part; LaVL, lateral amygdaloid nucleus, ventrolateral part; LaVM, lateral amygdaloid nucleus, ventromedial part.

**Figure 2 brainsci-07-00151-f002:**
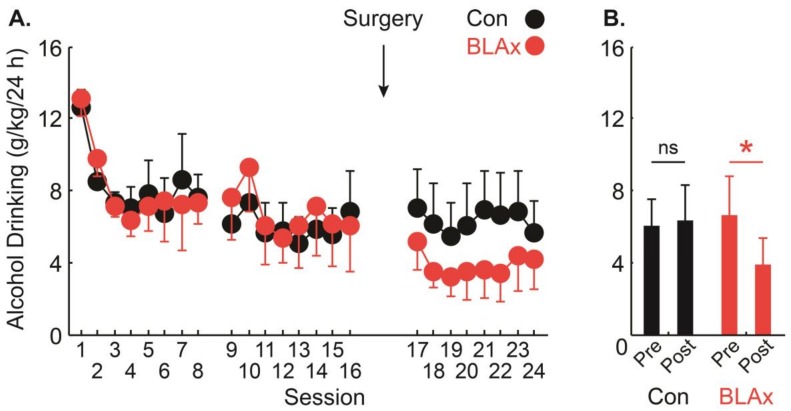
Alcohol drinking pre- and post-surgery. (**A**) Mean + SEM alcohol drinking (g/kg/24 h) for each voluntary access session is shown for control (Con—black) and mean − SEM for BLAx rats (red). Time of surgery is indicated by an arrow. (**B**) Mean + SEM alcohol drinking (g/kg/24 h) for the eight sessions prior to surgery (pre) and the eight sessions following surgery (post). Asterisks indicate the significance of the two-tailed, paired samples *t*-test (*p* < 0.05). NS indicates non-significance of the two-tailed, paired samples *t*-test (*p* > 0.1).

**Figure 3 brainsci-07-00151-f003:**
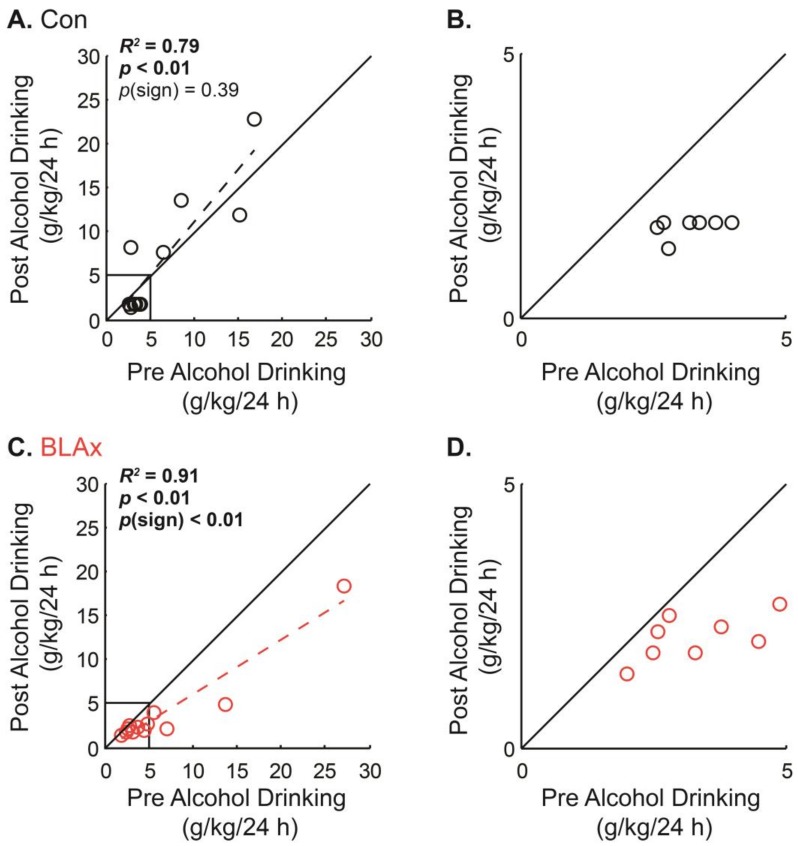
Relationship between alcohol drinking pre- and post-surgery. (**A**) Scatter plot compares pre-surgery drinking (mean of eight sessions prior to surgery) and post-surgery drinking (mean of eight sessions following surgery) for all control rats (Con). (**B**) Same data but zoomed in to visualize low drinking individuals. (**C**) Scatter plot compares pre-surgery drinking (mean of eight sessions prior to surgery) and post-surgery drinking (mean of eight sessions following surgery) for all BLAx rats. (**D**) Same data but zoomed in to visualize low drinking individuals. For each group, the square of Pearson’s correlation coefficient (*R*^2^) and its associated *p* value are reported. The *p*-value for a sign test comparing alcohol drinking (post – pre) to zero is reported. Note: scatter data points fully correspond to pre- and post-surgery alcohol drinking, shown in Figure 2B.

**Figure 4 brainsci-07-00151-f004:**
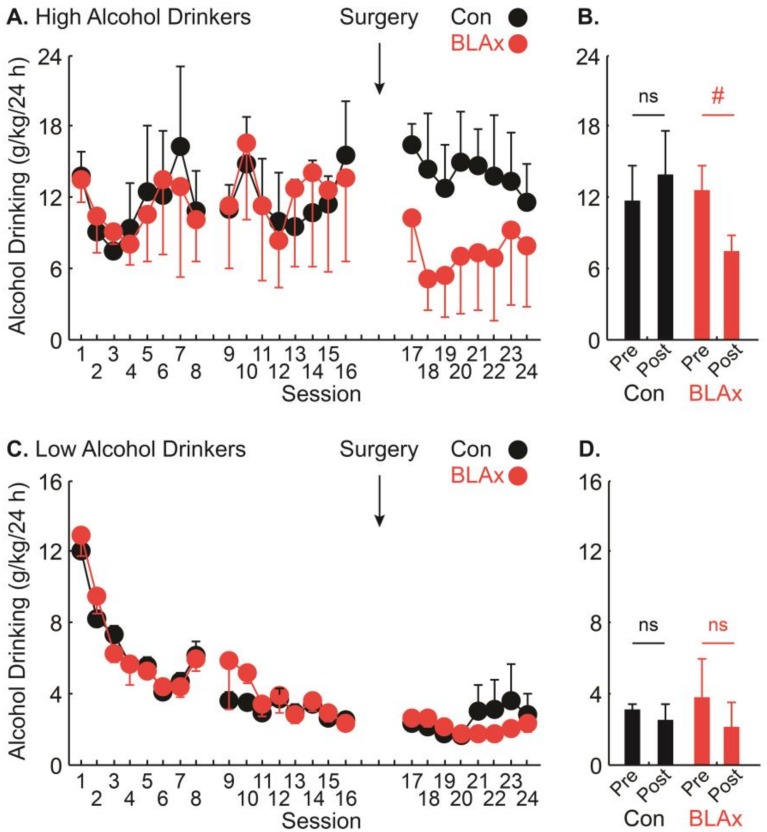
Alcohol drinking pre- and post-surgery for high and low drinkers. (**A**) Mean + SEM alcohol drinking (g/kg/24 h) for each voluntary access session is shown for the high drinking control (Con indicated in black) and mean − SEM shown for high drinking BLAx rats (red). The time of surgery is indicated by an arrow. (**B**) Mean + SEM alcohol drinking (g/kg/24 h) for the eight sessions prior to surgery (pre) and the eight sessions following surgery (post). (**C**) Mean + SEM alcohol drinking (g/kg/24 h) for each voluntary access session is shown for low drinking controls (Con indicated in black) and mean − SEM for low drinking BLAx rats (red). The time of surgery is indicated by an arrow. (**D**) Mean + SEM alcohol drinking (g/kg/24 h) for the eight sessions prior to surgery (pre) and the eight sessions following surgery (post). Pound sign indicates two-tailed, paired samples *t*-test (*p* = 0.09). NS indicates non-significance of the paired samples *t*-test (*p* > 0.1). Note: data are those exactly shown in Figure 2 (*n* = 12 per group), only separated with respect to high (*n* = 4 per group) or low (*n* = 8 per group) adolescent drinking.

**Figure 5 brainsci-07-00151-f005:**
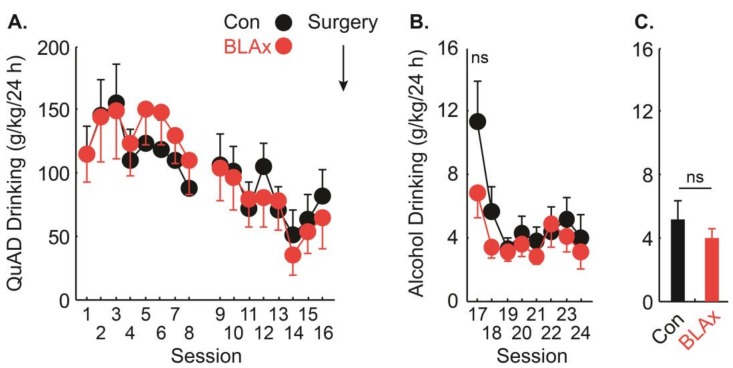
QuAD drinking pre-surgery and alcohol drinking post-surgery. (**A**) Mean ± SEM QuAD drinking (g/kg/24 h) for each voluntary access session prior to surgery is shown for controls (Con indicated in black) and BLAx rats (BLAx indicated in red). The time of surgery is indicated by an arrow. (**B**) Mean ± SEM alcohol drinking (g/kg/24 h) for each voluntary access session following surgery is shown for controls and BLAx rats. (**C**) Mean ± SEM alcohol drinking (g/kg/24 h) for the eight sessions following surgery (post). NS indicates non-significance of the two-tailed, independent samples *t*-test (*p* > 0.1).

**Figure 6 brainsci-07-00151-f006:**
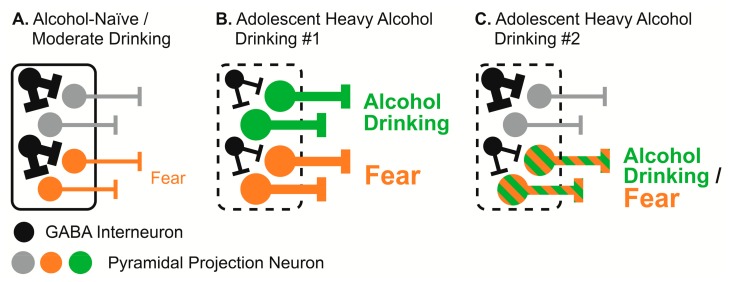
Working hypotheses for the alteration of BLA function by adolescent alcohol drinking. BLA is represented by rounded rectangles. Black circles indicate GABA interneurons, while all other circle colors (grey, orange, green) indicate pyramidal projection neurons. (**A**) In adult rats that are alcohol-naïve or were adolescent moderate alcohol drinkers, one class of pyramidal neuron signals fear (orange) while a second class is uninvolved in fear or alcohol (grey). GABA interneurons provide strong inhibitory input onto pyramidal neurons to modulate fear output, resulting in appropriate fear responses. (**B**) In scenario #1, adolescent heavy alcohol drinking recruits a new pyramidal population (green) whose activity is necessary to promote alcohol drinking. Simultaneously, adolescent drinking enhances the output of existing pyramidal neurons controlling fear (orange), and weakens GABAergic input onto each population. (**C**) In scenario #2, adolescent heavy alcohol drinking co-opts the pyramidal population controlling fear, whose activity now promotes alcohol drinking and fear. GABAergic input onto this population is simultaneously weakened. In both scenarios #1 and #2 the BLA is hyper-active, increasing fear and driving alcohol drinking.

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
