# Peer review of "Adolescent Alcohol Drinking Renders Adult Drinking BLA-Dependent: BLA Hyper-Activity as Contributor to Comorbid Alcohol Use Disorder and Anxiety Disorders"

_brainsci, 2017, doi:10.3390/brainsci7110151_

Round 1

Reviewer 1 Report

The manuscript by Moaddab and colleges investigates the role of the basolateral amygdala (BLA) in the effects of adolescent alcohol drinking on adult alcohol drinking in rats. The authors exposed four groups of animals to alcohol or an isocaloric control at adolescent age. Thereafter, animals received a lesion of the BLA or a sham lesion and were tested again for alcohol consumption at adult age. The original hypothesis – based on previous findings – was that adolescent alcohol exposure would enhance adult alcohol drinking and that this effect might be BLA dependent. In this study, the authors could not replicate previous findings that adolescent alcohol exposure enhances adult drinking. In fact, present data – even when torn apart – suggest that control groups without alcohol history drink more alcohol after the surgery (BLA lesion or sham lesion) than all alcohol pre-exposed animals. Since the replication of this effect is essential in this design, I think one cannot readily interpret present findings, even after the authors have reshaped their hypothesis and conclusions. More work is required to disentangle the conditions and mechanisms for the adolescent alcohol effect before publication can be considered.

Author Response

Reviewer #1

Thank you for your time and comments. Each is considered in turn below.

The manuscript by Moaddab and colleges investigates the role of the basolateral amygdala (BLA) in the effects of adolescent alcohol drinking on adult alcohol drinking in rats. The authors exposed four groups of animals to alcohol or an isocaloric control at adolescent age. Thereafter, animals received a lesion of the BLA or a sham lesion and were tested again for alcohol consumption at adult age. The original hypothesis – based on previous findings – was that adolescent alcohol exposure would enhance adult alcohol drinking and that this effect might be BLA dependent.

The description of the experimental design is accurate. However, the description of our specific hypothesis is slightly incorrect, which has implications for the comments below. Here is our original hypothesis as stated in the second paragraph of the introduction (Lines 36-38):

“Specifically, we hypothesize that alcohol drinking established in adolescence depends on the BLA in adulthood.”

We make no mention of adolescent alcohol drinking increasing adult alcohol drinking in our hypothesis. Even so, and as we will describe in greater detail below, we did find that in the sessions following surgery, highest alcohol drinking was observed in BLA-Intact rats given adolescent alcohol access. All other groups (QuAD-BLAx, QuAD-Control and EtOH-BLAx, showed numerically lower and equivalent drinking, especially in the final four sessions.

In this study, the authors could not replicate previous findings that adolescent alcohol exposure enhances adult drinking.

The finding that adolescent alcohol exposure enhances adult drinking is not as ubiquitous as one might assume. In the introduction, we cite a recent study by Amodeo and colleagues that convincingly shows that adolescent alcohol drinking can increase adult alcohol drinking. However, as we did in our 2015 paper (DiLeo et al), Amodeo et al go out of their way to discuss published studies that find no such effect of adolescent alcohol on adult drinking or even the opposite effect.

For your reference, here are studies finding no increase, or even a decrease, in adult alcohol drinking via adolescent alcohol drinking, as cited by Dileo et al and Amodeo et al:

M. Broadwater, E.I. Varlinskaya, L.P. Spear (2013) Effects of voluntary access to sweetened ethanol during adolescence on intake in adulthood. Alcoholism: Clinical and Experimental Research, 37, pp. 1048-1055.

J.P. Pian, J.R. Criado, B.M. Walker, C.L. Ehlers (2009) Milk consumption during adolescence decreases alcohol drinking in adulthood. Pharmacology, Biochemistry, and Behavior, 94, pp. 179-185.

S. Siegmund, V. Vengeliene, M.V. Singer, R. Spanagel (2005) Influence of age at drinking onset on long-term ethanol self-administration with deprivation and stress phases. Alcoholism: Clinical and Experimental Research, 29, pp. 1139-1145.

C.J. Slawecki, M. Betancourt (2002) Effects of adolescent ethanol exposure on ethanol consumption in adult rats. Alcohol, 26, pp. 23-30.

C.S. Vetter, T.L. Doremus-Fitzwater, L.P. Spear (2007) Time course of elevated ethanol intake in adolescent relative to adult rats under continuous, voluntary-access conditions. Alcoholism: Clinical and Experimental Research, 31, pp. 1159-1168.

In fact, present data – even when torn apart – suggest that control groups without alcohol history drink more alcohol after the surgery (BLA lesion or sham lesion) than all alcohol pre-exposed animals. Since the replication of this effect is essential in this design, I think one cannot readily interpret present findings, even after the authors have reshaped their hypothesis and conclusions.

This is not an accurate reading of the data. It is definitely the case that, in the very first alcohol access session, QuAD-Intact rats showed numerically high drinking. However, recall that prior to surgery, these rats were drinking large quantities of the control solution from the same bottle – roughly an order of magnitude greater than drinking exhibited by EtOH rats. Thus it is very likely that initial alcohol drinking in QuAD rats more reflected past drinking of a non-alcoholic solution. After the second post-surgery session, drinking dramatically decreased in QuAD-Intact rats and was low for the remainder of testing. Indeed, the highest post-surgery alcohol drinking (over all eight sessions) was found in EtOH-Intact rats – not in the either QuAD group. Indeed, if only the final four sessions are examined, the effect is even clearer.

                        8 Post-Surgery Sessions                 Final 4 sessions

EtOH-Intact:  6.33 ± 2.04 g/kg/24hr                         6.51 ± 2.09 g/kg/24hr

EtOH-BLAx:  3.85 ± 1.42 g/kg/24hr                         3.89 ± 1.70 g/kg/24hr

QuAD-Intact:             5.23 ± 1.12 g/kg/24hr                         4.31 ± 1.26 g/kg/24hr

QuAD-BLAx:             3.96 ± 0.60 g/kg/24hr                         3.73 ± 0.83 g/kg/24hr

In order to clarify our findings, we now report mean ± SEM drinking for the 8 post-surgery sessions (Line 163 and 128-129), as well as the final 4 sessions, in all groups (Lines 230-232).  

More work is required to disentangle the conditions and mechanisms for the adolescent alcohol effect before publication can be considered.

We certainly agree that more work is necessary to disentangle the conditions and mechanisms by which adolescent alcohol drinking alters alcohol drinking and behavior in adulthood. However, we also feel that the results of this study advance our understanding of the changes in neural function brought about by adolescent alcohol drinking. To our knowledge, the BLA has not previously been implicated as a target of adolescent drinking. That BLA lesions only decreased adult drinking in adolescents receiving alcohol access marks a significant advance.

Reviewer 2 Report

This manuscript assesses BLA involvement in alcohol drinking in adulthood following adolescent alcohol exposure in Long-Evans rats. Findings show that excitotoxic lesions to the BLA reduced adult alcohol drinking but not isocaloric quinine consumption. This effect was most pronounced in rats with high adolescent alcohol consumption. Overall, this manuscript is well written and the experiments are methodologically well conceived, with appropriate controls (isocaloric solution and sham microinfusions). There are a few minor corrections that could be made to improve the manuscript.

-        Methods

o   Under surgical procedures: Specify anesthesia and post-surgical analgesia used

o   A section detailing the general statistical analyses used should be included as a subsection under methods

-        Results:

o   Figs 2A, 4A, 4C, 5A, 5B: consistency in data reporting between groups would add clarity to the results. Specifically, error bars should be used consistently in both groups, such that Con and BLAx symbols include either mean+sem; mean-sem; or mean+/-sem. Thus, the figure captions as currently worded (stating mean +/- sem) are incorrect.

o   Fig 2B: it is not clear what comparison the asterisk represents, and though it seems to reference the difference between BLAx pre v. post, the overlap of error bars makes the significance questionable.   

o   If applicable, report if any animals were removed for surgical issues, lesions outside of the inclusion area, etc.

o   Additional histological details should be included such as the type of Nissl stain (e.g., cresyl violet, thionin) and methods pertaining to tissue sectioning

-        General

o   Update alcohol use disorders to alcohol use disorder (as per DSM-5 and NIAAA)

Author Response

Reviewer #3

This manuscript assesses BLA involvement in alcohol drinking in adulthood following adolescent alcohol exposure in Long-Evans rats. Findings show that excitotoxic lesions to the BLA reduced adult alcohol drinking but not isocaloric quinine consumption. This effect was most pronounced in rats with high adolescent alcohol consumption. Overall, this manuscript is well written and the experiments are methodologically well conceived, with appropriate controls (isocaloric solution and sham microinfusions). There are a few minor corrections that could be made to improve the manuscript.

Thank you for the kind words. Your concerns are addressed below.

-        Methods

o   Under surgical procedures: Specify anesthesia and post-surgical analgesia used

We have now specified that isoflurane was used for anesthesia and carprofen for post-operative analgesia. (Lines 84-85)

o   A section detailing the general statistical analyses used should be included as a subsection under methods

We apologize for this omission. The lack of a statistical analysis section was also pointed out by Reviewer 2. A detailed statistical section is now included. (Lines 96-102)

-        Results:

o   Figs 2A, 4A, 4C, 5A, 5B: consistency in data reporting between groups would add clarity to the results. Specifically, error bars should be used consistently in both groups, such that Con and BLAx symbols include either mean+sem; mean-sem; or mean+/-sem. Thus, the figure captions as currently worded (stating mean +/- sem) are incorrect.

Indeed, the captions were incorrect. We apologize and have fixed the mistakes. Now each figure caption appropriately describes the error bar direction associated with each group/graph. 

o   Fig 2B: it is not clear what comparison the asterisk represents, and though it seems to reference the difference between BLAx pre v. post, the overlap of error bars makes the significance questionable.  

The stated comparison is correct. Significance is for a within-subjects t-test, pre-surgery vs. post-surgery drinking. We have edited the figure to make the comparison more obvious.

o   If applicable, report if any animals were removed for surgical issues, lesions outside of the inclusion area, etc.

This was also brought up by Reviewer 2. To be sure, there was variability in lesion size across individuals. However, each BLAx rat had evidence of neurotoxic damage. Indeed, our lesion procedure was designed to error on the side of producing damage – as there are two injection locations per hemisphere, for a total of four injections site per BLAx individual.

o   Additional histological details should be included such as the type of Nissl stain (e.g., cresyl violet, thionin) and methods pertaining to tissue sectioning

We apologize for the omission. This was an oversight. We now include all methods pertaining to histology and staining. (Lines 91-94)

-        General

o   Update alcohol use disorders to alcohol use disorder (as per DSM-5 and NIAAA)

We have updated all mention of alcohol use disorder to reflect the DSM-5 and NIAAA.

Reviewer 3 Report

In this manuscript by Moaddab et al., the authors investigate, through excitotoxic lesion of the basolateral amygdala (BLA), whether this brain region is responsible for continued alcohol drinking in adulthood after exposure to voluntary drinking during adolescence. Their findings indicate that, in BLA-lesioned rats, continued drinking in adulthood is reduced, particularly in rats that had been heavy drinkers in adolescence. However, the BLA does not appear to be necessary for alcohol consumption, as was shown in animals that were not exposed to alcohol in adolescence. The authors propose two possible mechanisms by which heavy adolescent drinking leads to hyper-excitability of alcohol- and rapid fear-related neurons within the BLA.

-          Some of the comparisons being made were difficult to follow, partially due to a lack of specificity in what type of statistical test is being run. A statistics section in the methods is necessary to help clarify these issues.

-          In the Methods section, there were two separate sets of surgical coordinates given with separate injection volumes associated with each (lines 76-77). It should be clarified whether both of these sets of coordinates/volumes were used, and if so, what determined the use of one over the other.

-          What is the rationale/justification for exposing animals to alcohol until P56?

-          Although it is not stated, I assume that not every surgery successfully targeted the BLA. If not, what does the data from those animals look like? It could serve as another control for the role of the BLA in modulating alcohol consumption in animals with a history of adolescent alcohol exposure.

-          One of the conclusions of this study is that the BLA is involved in drinking following a history of heavy adolescent alcohol exposure (since the effect was only seen in high drinkers). However, it is interesting that the levels did not drop to the levels of low drinkers. This should be discussed.

-          Related to the last comment, were the animals video taped during the drinking sessions? If so, it would be interesting to see the patterns of drinking in high vs low drinkers (i.e. do high drinkers drink during the whole session or do they binge drink and then stop?).

-          While the flow of logic from one subsection the next in the Results section can be inferred, a statement explicitly stating how the results from one section/experiment led to the development of the next would be useful for confirming the assumption of the reader.   

-          Since part of the conclusions from this study are based on a comparison to this group’s previous work (DiLeo et al., 2016), it would be beneficial to describe the last study and how it fits into the context of the current study. This would help understand Figure 6 (proposed mechanism(s)), which was very difficult to interpret. Please consider revising the figure itself and the associated text for clarity.

-          In the Results section, it is not explicitly stated why only the last 8 sessions of pre-surgery drinking and the first 8 sessions of post-surgery drinking were included in the data analysis. Please include a statement about this. 

-          There must be a typographical error on the scale for the y axis for Figure 5A, as this would calculate to animals drinking close to 1 liter of ethanol solution in a 24 hour period.

-          The type of graph chosen for Figures 2B, 4B, and 4D do not seem sufficient for accurately representing the data/results. For example, in Fig. 2B there was a significant effect, while in Fig. 4B only a trend was reported. However, visually, it looks as though the opposite would be true. One suggestion for making this clearer is to superimpose individual data points over the bars to demonstrate variability and to see the change in alcohol consumption in individuals before and after surgery.

-          In general, the Discussion section is lacking a discussion of the results obtained in the present study and how they support/contradict existing literature. This is an essential component and needs to be included. As an example, what would happen if the BLA was taken offline before alcohol consumption commenced in adolescents?

-          This document needs to be thoroughly proof-read for grammatical mistakes and spelling errors.

Author Response

Reviewer #2

In this manuscript by Moaddab et al., the authors investigate, through excitotoxic lesion of the basolateral amygdala (BLA), whether this brain region is responsible for continued alcohol drinking in adulthood after exposure to voluntary drinking during adolescence. Their findings indicate that, in BLA-lesioned rats, continued drinking in adulthood is reduced, particularly in rats that had been heavy drinkers in adolescence. However, the BLA does not appear to be necessary for alcohol consumption, as was shown in animals that were not exposed to alcohol in adolescence. The authors propose two possible mechanisms by which heavy adolescent drinking leads to hyper-excitability of alcohol- and rapid fear-related neurons within the BLA.

This is an excellent summary of our findings. Thank you for your time and effort.

-          Some of the comparisons being made were difficult to follow, partially due to a lack of specificity in what type of statistical test is being run. A statistics section in the methods is necessary to help clarify these issues.

We apologize for the omission of a statistics section. This was a complete oversight. A statistics section is now included and all analyses are described in full. (Line 95)

-          In the Methods section, there were two separate sets of surgical coordinates given with separate injection volumes associated with each (lines 76-77). It should be clarified whether both of these sets of coordinates/volumes were used, and if so, what determined the use of one over the other.

In this study, each BLAx rats received a total of 4 infusions. In each hemisphere, one injection was given more ventrally, and a second injection was given more dorsally. This was done in order to more consistently damage the full dorsal-ventral extent of the BLA. This is now clarified in the manuscript and the logic for the 4 total infusions (2 per hemisphere) is explained. (Lines 79-83)

-          What is the rationale/justification for exposing animals to alcohol until P56?

This is a good question. The goal of this experiment (and previous adolescent drinking experiments from our lab) is to maximize exposure to voluntary alcohol access in adolescence. We use a procedure developed by Simms and colleagues (2008) that gives 24-hr access three times a week. Because rats only receive access 3 times a week, many weeks of access are necessary in order to provide sufficient experience. Our rats receive a total of 16 voluntary drinking sessions. Additionally, we wish to maximize the adolescent period over which rats receive access. There is not a consensus view of the age reach adulthood. Depending on the criteria (size, reproductive status, brain development) ages range from 60 to 90 days old. However, we have found 60 to be absolute lowest. By stopping at day 56 ± 3 we are ensuring – to the best of our ability – that alcohol access was restricted to adolescence. We now state this rationale in the methods and provide a citation. (Lines 64-67)

-          Although it is not stated, I assume that not every surgery successfully targeted the BLA. If not, what does the data from those animals look like? It could serve as another control for the role of the BLA in modulating alcohol consumption in animals with a history of adolescent alcohol exposure.

To be sure, there was variability in lesion size across individuals. However, each BLAx rat had evidence of neurotoxic damage. Indeed, our lesion procedure was designed to error on the side of producing damage – as there are two injection locations per hemisphere.

-          One of the conclusions of this study is that the BLA is involved in drinking following a history of heavy adolescent alcohol exposure (since the effect was only seen in high drinkers). However, it is interesting that the levels did not drop to the levels of low drinkers. This should be discussed.

We agree this is interesting and we now discuss this observation. In a nutshell, we do not think the BLA is the only structure necessary for adult drinking established in adolescence. Rather, the BLA is a key node in a larger network. Removing the BLA reduces drinking but does not do so completely. (Lines 235-249)

-          Related to the last comment, were the animals video taped during the drinking sessions? If so, it would be interesting to see the patterns of drinking in high vs low drinkers (i.e. do high drinkers drink during the whole session or do they binge drink and then stop?).

The animals were not videotaped during the drinking session. Also, no lick data were recorded. We use standard ball bearing sipper tubes and do not have lickometers. Currently, we are only able to measure total drinking in a 24 hour period. However, we are very interested in looking at the pattern of licking/drinking and how this pattern may not only differ between low and high drinkers, but that there might also be different types of high drinkers (binge vs steady). We have been made aware of capacitance-based lick detectors that may do exactly what we want:

http://www.noldus.com/phenotyper/lickometer

We are in contact with the company and are getting information about the product’s capability and pricing.

-          While the flow of logic from one subsection the next in the Results section can be inferred, a statement explicitly stating how the results from one section/experiment led to the development of the next would be useful for confirming the assumption of the reader.  

We agree the transition between each set of results was not fluid. We have now revised the results section to better transition between the analyses.

-          Since part of the conclusions from this study are based on a comparison to this group’s previous work (DiLeo et al., 2016), it would be beneficial to describe the last study and how it fits into the context of the current study. This would help understand Figure 6 (proposed mechanism(s)), which was very difficult to interpret. Please consider revising the figure itself and the associated text for clarity.

We now spend a paragraph of the discussion on the results of the DiLeo study and have integrated those results with our proposed mechanisms in figure 6. (Lines 276-292)

-          In the Results section, it is not explicitly stated why only the last 8 sessions of pre-surgery drinking and the first 8 sessions of post-surgery drinking were included in the data analysis. Please include a statement about this.

We have now included this statement in the results section:

“In order to determine the effect of BLA lesion on alcohol drinking established in adolescence, we analyzed the eight voluntary sessions prior to and following surgery. Pre-surgery drinking had stabilized during these eight sessions, as indicated by ANOVA revealing no main effect of session. Further, examining the eight sessions on either side of surgery assured that analysis of drinking was balanced (identical number of pre-surgery and post-surgery sessions) and complete (analyzing the full post-surgery drinking data collected).” (Lines 150-155)

-          There must be a typographical error on the scale for the y axis for Figure 5A, as this would calculate to animals drinking close to 1 liter of ethanol solution in a 24 hour period.

We have checked the y axis scale and it is correct. We think the issue may be with our incomplete description of our dependent measure. The measure of drinking we report is:

grams (solution consumed) / kg (body weight) per 24 hours (total time given access)

abbreviated as: g/kg/24hr

Below we show g/kg/24hr drinking levels based on hypothetical data from a range of drinking and body weights:

Solution Consumed (g)

Body Weight (kg)

g/kg/24hr

5

0.1

50.0

5

0.3

16.7

10

0.1

100.0

10

0.3

33.3

50

0.1

500.0

50

0.3

166.7

As expected, low drinking (5 g) with high body weight (0.3 kg) yields the lowest drinking in g/kg/24hr (16.7 k/kg/24hr) where as high drinking (50 g) with low body weight (0.1 kg) yields the highest drinking in g/kg/24hr (500 g/kg/24hr).

To avoid confusion, we have provided more detail about the dependent measure in the methods section (Lines 72-74):

“Experimental bottles were weighed (g) prior to, and following, 24-hr access to solutions. Rats were weighed (kg) immediately following experimental bottle removal. Drinking is reported in grams (solution consumed)/kg (body weight) per 24 hours (total time given access): g/kg/24 hr.”

-          The type of graph chosen for Figures 2B, 4B, and 4D do not seem sufficient for accurately representing the data/results. For example, in Fig. 2B there was a significant effect, while in Fig. 4B only a trend was reported. However, visually, it looks as though the opposite would be true. One suggestion for making this clearer is to superimpose individual data points over the bars to demonstrate variability and to see the change in alcohol consumption in individuals before and after surgery.

We agree it is odd that the comparison in Figure 2B returns a significant t-test result, yet Figure 4 B does not. However, it is entirely due to sample size. Figure 2B compared drinking for all 12 BLAx rats and all 12 decreased drinking following surgery. Figure 4B only compared drinking for the top 4 BLAx rats. Because of the small sample size, significance was only approached. We agree that visualization of the individual data points is helpful, and these are the data that comprise the scatter plots in Figure 3. We thought it redundant to include the individual data in Figures 2 and 4 as well. We now note in the Figure 3 caption, that scatter data points fully correspond to pre- and post-surgery alcohol drinking shown in Figure 2B. Further, we explicitly state that Figure 4 data are exactly as in Figure 2; only Figure 2 plotted all individuals in each group (n=12) whereas Figure 4 separated High (n=4) and Low (n=8) adolescent drinkers.

-          In general, the Discussion section is lacking a discussion of the results obtained in the present study and how they support/contradict existing literature. This is an essential component and needs to be included. As an example, what would happen if the BLA was taken offline before alcohol consumption commenced in adolescents?

We agree this was lacking. This section of the discussion has been expanded. (Lines 262-275)

-          This document needs to be thoroughly proof-read for grammatical mistakes and spelling errors.

We apologize for the errors and have thoroughly checked the manuscript for grammar and other mistakes.

Round 2

Reviewer 3 Report

The authors addressed most of my concerns. However, the calculation used for ethanol consumption (grams/kilograms) is not correct as the raw weight of volume consumed by the animal is not actually grams of ethanol. The authors needs to convert this weight of volume (i.e. the bottle weighed 5 grams less at the end of the session compared to the beginning) to actual grams of ethanol, considering that 20% ethanol was used. Whether correcting these calculations  will change the data is left to be determined. Based on this, the results are still inconclusive and uninterpretable data.

Author Response

We apologize for overlooking this concern in the first review. Our primary behavioral measure is alcohol drinking, which refers to drinking of the 20% ethanol solution. Of course, the level of ethanol drinking must be lower, due simply to the fact that our alcohol solution is ~80% water.

Critically, converting alcohol drinking to ethanol drinking results in identical statistical results for every analysis performed in this manuscript. This is because converting alcohol drinking to ethanol drinking simply requires multiplying alcohol drinking by a constant factor (0.162), in order to determine g of ethanol per mL of alcohol. Thus, the underlying distribution of drinking is identical for alcohol and ethanol drinking.

In order to make this clear to the reader, we now include a supplementary figure in which we plot mean and SEM ethanol drinking in the exact same manner as in Figure 2A, only adjusting the y-axis scale. Evident from the figure, completely identical patterns are observed for alcohol and ethanol drinking. As would be expected, ANOVA results for are also identical for the two measures. To further demonstrate that these measures are identical, we plot pre vs post ethanol drinking for Controls (Supplemental Figure B) and BLAx (Supplemental Figure C), and we observe identical relationships as show in Figure 3. Finally, we compare alcohol and ethanol drinking for each group, for pre-surgery and post-surgery drinking, and demonstrate that each comparison yields R2 values of 1.00 (Supplemental Figure D-G). The two measures are perfectly correlated and are therefore identical.

The Dependent Measures section of the Results has been expanded to point out that ethanol drinking can be derived from alcohol drinking and that ethanol drinking levels are lower than that of alcohol. The supplemental figure is referenced in this section, to point out that these two measures yield identical statistical results. 
